# Effects of Preparation Method on the Physicochemical Properties of Cationic Nanocellulose and Starch Nanocomposites

**DOI:** 10.3390/nano9121702

**Published:** 2019-11-28

**Authors:** Lina Han, Wentao Wang, Rui Zhang, Haizhou Dong, Jingyuan Liu, Lingrang Kong, Hanxue Hou

**Affiliations:** 1College of Food Science and Engineering, Shandong Agricultural University, Tai’an 271018, China; hanlina1994@126.com (L.H.); wwtlxm@126.com (W.W.); xuyuyisha@163.com (R.Z.); hzhdong@sdau.edu.cn (H.D.); LJY1664961793@126.com (J.L.); 2State Key Laboratory of Biobased Material and Green Papermaking, Qilu University of Technology, Shandong Academy of Sciences, Jinan 250000, China; 3College of Agronomy, Shandong Agricultural University, Tai’an 271018, China

**Keywords:** nanocellulose, cationic microcrystalline cellulose, high-intensity ultrasonication, high-pressure homogenization, acid hydrolysis, starch nanocomposite films

## Abstract

Nanocellulose (NC) has attracted attention in recent years for the advantages offered by its unique characteristics. In this study, the effects of the preparation method on the properties of starch films were investigated by preparing NC from cationic-modified microcrystalline cellulose (MD-MCC) using three methods: Acid hydrolysis (AH), high-pressure homogenization (HH), and high-intensity ultrasonication (US). When MD-MCC was used as the starting material, the yield of NC dramatically increased compared to the NC yield obtained from unmodified MCC and the increased zeta potential improved its suspension stability in water. The NC prepared by the different methods had a range of particle sizes and exhibited needle-like structures with high aspect ratios. Fourier transform infrared (FTIR) spectra indicated that trimethyl quaternary ammonium salt groups were introduced to the cellulose backbone during etherification. AH-NC had a much lower maximum decomposition temperature (T_max_) than HH-NC or US-NC. The starch/HH-NC film exhibited the best water vapor barrier properties because the HH-NC particles were well-dispersed in the starch matrix, as demonstrated by the surface morphology of the film. Our results suggest that cationic NC is a promising reinforcing agent for the development of starch-based biodegradable food-packaging materials.

## 1. Introduction 

Cellulose is the most abundant renewable natural resource and thus has attracted the interest of researchers around the world [1]. Native cellulose consists of amorphous and crystalline regions; when subjected to appropriate treatments such as mechanical, chemical, or enzymatic methods; nanocellulose (NC) can be obtained by breaking down amorphous regions [2]. In recent years; NC has become one of the most promising nanomaterials and has attracted increasing attention in the field of nanocomposites because of its appealing intrinsic properties which include its high specific surface area; high aspect ratio; low density; high chemical reactivity; high tensile strength; and high Young’s modulus [3,4,5,6]. 

Multiple technologies have been developed to prepare NC from cellulosic materials, including ultrasonication [7], high-pressure homogenization [8], steam explosion treatment [9], high-speed grinding [10], acid hydrolysis [11], TEMPO (2,2,6,6-tetramethylpiperidine-1-oxyl radical) oxidation [12], and combined treatments [13]. The morphology and properties of NC particles determine their application properties and are affected by preparation methods [14]. Recently, there has been increasing interest in the production of NC from microcrystalline cellulose (MCC) because MCC has a high cellulose content, which is affected by its pectin, hemicellulose, lignin, and other lignocellulosic components [15]. The main drawback of acid hydrolysis, ultrasonication, and high-pressure homogenization methods using MCC is their low NC yield [16,17,18]. Several studies have attempted to increase the NC yield from MCC; however, their methods were limited by complicated pretreatment steps, high-energy consumption, long treatment times, and high water consumption [18,19,20]. 

Nanocellulose has been widely used as a reinforcing agent for various advanced composite applications because of its unique advantages [21], which include improving the mechanical and water-vapor barrier properties of starch films [22]. However, NC particles tend to agglomerate via van der Waals forces and hydrogen bonding during film preparation because a large number of hydroxyl groups are present on the NC surface, which restricts the advantages they offer. The dispersion of NC in starch matrices and the interfacial adhesion between NC and starch are widely accepted as the critical factors determining the reinforcement effect of NC in starch [14]; the advantages offered by NC—i.e., large aspect ratio, high modulus, and large surface area to interact with the starch matrix—can only be fully realized when NC is homogeneously distributed in the starch matrix. 

Surface modification of NC to improve its dispersion and compatibility with polymer matrices has been widely studied and different surface modification methods have been reported, such as etherification [23], esterification [24], ionic interaction [25], silylation [26], and oxidation [27]. Most of the previous studies modified the NC directly. However, surface modification of NC is complicated and has limited effectiveness because of the highly aggregated structure of NC [28]. Therefore, it was difficult to prepare NC with a modified group. Alkaline solutions are known to swell cellulose samples. Alkali-swelling can disrupt hydrogen bonding between microfibrils in pulp fibers and could facilitate nanofibrillation [29]. This suggests that pre-swelling can be conducive to nanofibrillation of the MCC. This has positive significance for resource conservation and environmental protection. Thus, the preparation of NC from previously swelling and modified MCC is a new strategy that warrants study. 

In order to efficiently obtain modified NC with high dispersibility in starch film, a new preparation process has been developed. Microcrystalline cellulose was swelling by sodium hydroxide solution and first modified with 3-chloro-2-hydroxypropyl trimethylammonium chloride. Then, cationic NCs were prepared from previously modified MCC by acid hydrolysis, high-pressure homogenization, and high-intensity ultrasonication. The yield, zeta potential, average particle size, dispersion stability, morphology, crystallinity, chemical structure, and thermal stability of the NC particles were studied. Finally, the effects of the NC preparation methods on the physico-chemical properties of starch nanocomposites were investigated. 

## 2. Materials and Methods 

### 2.1. Materials

Cotton microcrystalline cellulose (MCC) was purchased from Huzhou City Linghu Xinwang Chemical Co., Ltd. (Huzhou, China), (3-chloro-2-hydroxypropyl)trimethylammonium chloride (CHPTA) was obtained from Chengdu Aikeda Chemical Reagent Co., Ltd. (Chengdu, China), and starch was purchased from Hangzhou Starpro Co., Ltd. (Hangzhou, China). 

### 2.2. Cationic Modification of Microcrystalline Cellulose (MCC)

Microcrystalline cellulose was soaked in a 10% sodium hydroxide solution with a solid to liquid ratio of 1 to 10 (1 g/10 mL) at 25 °C for 24 h, whose purpose was to swell cellulose so that CHPTA could enter the cellulose. Then the MCC suspension was centrifuged at 3000 rpm for 10 min and neutralized with diluted hydrochloric acid. After centrifugation, the MCC was dried in an oven at 60 °C for 48 h and subsequently ground into powder. 

A 30 g sample of the MCC powder was dispersed in 600 mL of deionized water and 9.87 g sodium hydroxide was added while stirring at room temperature for 30 min. The cationic etherifying agent was then gradually added with continuous stirring and the molar ratio of sodium hydroxide to CHPTA was fixed at 1.2:1. The reaction mixture was stirred for 5 h at 65 °C and the resultant suspension was centrifuged at 5000 rpm for 10 min to obtain the precipitate. The precipitate was suspended in deionized water to remove CHPTA, then dried and ground into powder. The modified MCC is referred to as MD-MCC.

### 2.3. Preparation of Nanocellulose (NC)

For acid hydrolysis, 10 g of the MCC/MD-MCC was dispersed in 100 mL of 60% (*v*/*v*) sulfuric acid. Hydrolysis was conducted at 55 °C with constant stirring for 1 h, after which the reaction was stopped by adding cold water (10-fold dilution). The suspension was centrifuged at 5000 rpm for 15 min and dialyzed with distilled water for several days until the dialysate became neutral; the neutral suspension was centrifuged at 10,000 rpm for 10 min to recover the colloidal suspension. The resultant NC is referred to as AH-NC.

For high-pressure homogenization, 1 g of MCC/MD-MCC was added to 100 mL of deionized water and homogenized 4 times at 800 bar using a SCIENTZ-150 high pressure homogenizer (Ningbo Xinzhi Biotechnology Co., Ltd., Ningbo, China). The resultant cellulose suspension was centrifuged at 10,000 rpm for 5 min to recover the precipitate, which was then suspended in deionized water. This process was repeated 3 times. The NC colloidal suspension obtained as the supernatant is referred to as HH-NC.

For high-intensity ultrasonication, MCC/MD-MCC was soaked in deionized water for 24 h with a solid to liquid ratio of 1:100. The suspension was homogenized using the TJS-3000 ultrasonicator (1750 W for 30 min) and subsequently centrifuged at 10,000 rpm for 5 min to recover the precipitate, which was suspended in deionized water. This process was repeated 3 times. The NC colloidal suspension obtained as the supernatant is referred to as US-NC.

The yield of NC was calculated gravimetrically according to Equation (1): (1)Yield % = Weight of NCWeight of MCC ×100%

### 2.4. Preparation of Starch/NC Composite Films

Starch/NC composite films were prepared using a solution casting method. A certain amount of NC (5% w/w of starch) was dispersed in 100 mL of deionized water by ultrasonication at 600 W for 5 min; 3 g of starch and 0.9 g of glycerol were subsequently added and the suspension was stirred at 85 °C for 1 h. The sample solution was poured onto a PTFE glass plate (24 cm × 12 cm), dried at 60 °C for 6 h, and then peeled off and kept at 23 °C and 53% relative humidity for at least 7 d prior to testing.

### 2.5. Characterization of MCC and NC

#### 2.5.1. Particle Size and Zeta Potential

The average particle size and zeta potential of the NC in aqueous suspensions were determined using a Nanobrook ZetaPlus Potential Analyzer (Brookhaven Instruments Corporation, Holtsville, NY, USA) under the following conditions: 1.3328 water refractive index, 90° angle, and 25 °C. 

#### 2.5.2. Scanning Electron Microscopy (SEM)

The morphology of the MCC and MD-MCC were analyzed by scanning electron microscope (QUANTA FEG 250, FEI, Hillsboro, OR, USA) at a voltage of 5.0 kV. The specimens were placed on a bronze stub and sputter-coated with gold before testing.

The morphological characteristics of NC were studied by transmission electron microscopy (TEM) with a TECNAI 20 U-TWIN microscope (PHIA, Eindhoven, Netherlands) using an acceleration voltage of 100 kV. The prepared suspension was spotted on to a carbon coated copper grid. The grid was dried before TEM analysis. The length and diameter of NC were measured using image analysis (Nano Measure software) at least 100 randomly selected NC fibrils in certain TEM images.

#### 2.5.3. X-ray Diffraction (XRD)

X-ray diffraction (XRD) analysis of the NC samples were determined with a D8 X-ray diffractometer (Bruker-AXS, Karlsruhe, Germany) equipped with a copper target (λ = 0.15406 nm) at 30 mA and 40 kV. Data were recorded in the range (2θ) of 5–40° at a scan rate of 0.02°·s^–1^. The crystallinity index was calculated with Equation (2):(2)$$\mathrm{CrI}\;=\;\frac{I_{200}-\;I_{am}}{I_{200}}$$ where *CrI* is the crystallinity index, *I*_200_ is the maximum intensity of the diffraction from the 200 plane, and *I_am_* is the intensity of particles scattered by the amorphous part of the sample. 

#### 2.5.4. Fourier Transform Infrared (FTIR) Spectroscopy

The fourier transform infrared (FTIR) spectra of MCC, MD-MCC and NC were recorded using a Nicolet iS5 spectrometer with iD5 ATR sampling accessory (Thermo Fisher Scientific, Waltham, MA, USA). All spectra were collected from an accumulation of 32 scans at wavelengths ranging between 4000 and 600 cm^−1^. 

#### 2.5.5. Thermogravimetric Analysis (TGA)

Thermogravimetric analysis (TGA) was performed to analyze the thermal properties of NC samples using a TA-60 thermogravimetric analyzer (SHIMADZU, Japan) under nitrogen flow of 50 mL/min. Heating temperature ranged from 25 °C to 500 °C at a heating rate of 10 °C·min^−1^. 

### 2.6. Characterization of Starch/NC Composite Films

#### 2.6.1. Mechanical Properties 

Tensile strength (TS, MPa), elongation at break (EAB, %) and elastic modulus (EM, GPa) of the films were measured with an XLW auto tensile tester (Labthink Instruments Co. Ltd., Jinan, China) according to ASTM (American Society of Testing Materials) D882-12 (2012). The samples were cut into strips (dimensions, 150 mm × 15 mm). The initial distance between the grips was 100 mm and the test speed was set at 100 mm/min. Each test was repeated at least six times. 

#### 2.6.2. Water Vapor Permeability (WVP)

Water vapor permeability (WVP) was measured according to ASTM E96/E96M-16 (2016) using a PERME™ W3/030 automatic water vapor permeability tester (Labthink Instruments Co., Ltd., Jinan, China). Films were cut into round specimens (80 mm in diameter) with a special sampler. The test was performed at 38.0 °C and 90% RH (relative humidity) with a preheating time of 4 h and a weighing interval of 120 min. The WVP of each sample was obtained from the average of three measurements. 

#### 2.6.3. Morphology of Starch Nanocomposite Films 

The morphology of starch composite films was analyzed by scanning electron microscope (QUANTA FEG 250, FEI, Hillsborough, OR, USA) at a voltage of 5.0 kV. The specimens were placed on a bronze stub and sputter-coated with gold before testing.

### 2.7. Statistical Analysis 

Statistical differences between the properties of MCC, MD-MCC, NC, and starch films were determined by analysis of variance (ANOVA) via SPSS 21 (IBM Co., Armonk, NY, USA). Comparisons of mean values were performed using Duncan’s multiple range tests (*p* < 0.05).

## 3. Results and Discussion 

### 3.1. Effect of Preparation Method on the Yield of NC Prepared from MCC and Modified Microcrystalline Cellulose (MD-MCC)

Low NC yield is a limiting factor in industrial production and commercial applications. The yields of NC prepared from MCC and MD-MCC acid hydrolysis, high-pressure homogenization, and high-intensity ultrasonication are shown in Table 1. The yield of AH-NC prepared from MD-MCC was 30.63% which increased by 48.1%, compared with that prepared from MCC. Moreover, the yields of HH-NC and US-NC prepared from MCC were only 2.04% and 3.57%, respectively, whereas the yields from MD-MCC were 33.08% and 14.18%; these correspond to increases of approximately 16.2 and 3.97 times, respectively. These results demonstrate that cationic etherification of MCC could significantly increase the NC yield of different preparation methods, especially high-pressure homogenization and high-intensity ultrasonication. The NC yields of AH-NC and HH-NC prepared from MD-MCC are comparable to other reports of NC obtained under the similar conditions [30,31], and it can be increased with the optimization of the treatment conditions; however, there is minimal information on the effect of separate ultrasound processing on NC yield as ultrasonication has primarily been used as an auxiliary processing method.

Natural cellulose is insoluble and tends to agglomerate in water as it forms an extensive network of intermolecular and intramolecular hydrogen bonds, which can clog valves in homogenizers. Thus, the HH yield of NC prepared from MCC is typically low [32]. During the preparation of US-NC, ultrasonication affected the surface to the inner amorphous regions of MCC and caused MCC to break into submicron fragments instead of directly forming NC. However, the small size of the MCC fragments impeded the ultrasonication process and resulted in the low NC yield [12,18]. Conversely, acid molecules can rapidly penetrate into the inner amorphous regions of the cellulose fibrils where they disintegrate amorphous regions, reduce the size of the cellulose fibers, and ultimately release cellulose nanofibrils [33]; thus, AH-NC had a higher yield than HH-NC or US-NC. Considering these processes, cationic modification of MCC can substantially affect NC preparation by acid hydrolysis, high-pressure homogenization, and high-intensity ultrasonication methods. 

### 3.2. Effect of Preparation Method on the Suspension Stability of NC Prepared from MCC and MD-MCC

The suspension states of NC prepared from MCC and MD-MCC by acid hydrolysis, high-pressure homogenization, and high-intensity ultrasonication are shown in Figure 1. All NC samples were well-dispersed in water and their suspensions were stable and uniform without any stratification when fresh. After 3 days at 20 °C, flocculation and precipitation occurred in the HH-NC and US-NC samples prepared from MCC whereas the other NC samples remained homogeneous, which indicates that the NC particles prepared by MD-MCC were relatively stable in water. 

Net charge is a critical characteristic that affects the stability of NC particles. In general, higher absolute values of zeta potential correspond to better dispersion and stability [2]. The results from this study reveal that the introduction of trimethyl quaternary ammonium groups by cationic modification of MCC (confirmed by the FTIR results) helped to increase the NC zeta potential and further improved the stability and dispersion of the NC particles. Based on the measured yield and application properties, the NC samples prepared from MD-MCC were selected for further analysis. 

### 3.3. NC Morphology

The SEM micrographs of MCC and MD-MCC shown in Figure 2A reveal that MCC particles had irregular shapes with different dimensions and MD-MCC particles were swollen and porous with a rough surface. Moreover, cationic modification damaged MCC particles and eroded their surface such that the outer layer of the fibers was disrupted and cracked along the inner structure, exposing the fibril strand. These changes to the MCC granular structures facilitated the substantial increase in NC yield as discussed above. 

The TEM micrographs of AH-NC, HH-NC, and US-NC shown in Figure 2B illustrate their distinctive morphologies and dispersion states. Small bundles of needle-like cellulose fibers with nanoscale diameters were observed for AH-NC and branches of smaller bundles or partly individualized nanofibers were attached to the aggregates as well (arrows in Figure 2B(c)). The formation of such aggregates significantly reduced the surface area of the AH-NC particles and thus hindered their reinforcing ability. Conversely, the HH-NC and US-NC nanofibers were well-dispersed and more individualized. 

The NC dispersion state was directly associated with the surface interactions between adjacent NC particles because different surface interactions exist in aqueous solution, including attraction forces (e.g., hydrogen bonding) and repulsion forces (e.g., electrostatic repulsion). Attraction and repulsion forces are expected to compete with each other and thus determine the distinctive dispersion state of aqueous NC from different preparation methods. Therefore, the uniform dispersion of HH-NC and US-NC in aqueous solution was ascribed to profound repulsion forces whereas the AH-NC aggregates were attributed to predominant attraction forces. These theories were confirmed by the zeta potential values shown in Table 1. As expected, the order of absolute zeta potential values of NC prepared from MD-MCC was AH-NC < HH-NC < US-NC, which corresponds to the observed dispersion states. Because of the sulfate anions present during acid hydrolysis, the zeta potential value of AH-NC was not substantially changed. The dispersion states also indicate that hydrogen bonding between nanofibers was lower for HH-NC and US-NC.

### 3.4. Length-Frequency and Diameter-Frequency Histograms

Length-frequency and diameter-frequency histograms (Figure 3) were prepared using the TEM data. Aggregation and overlapping nanofibers make it difficult to accurately measure dimensions; thus, only individual nanofibers with clearly identifiable ends were measured. The AH-NC exhibited a wide distribution of lengths whereas the diameter distribution had a narrower range of 2 to 8 nm with the maximum at 4.5 nm. The length distributions of HH-NC and US-NC were 25 to 275 nm and 75 to 350 nm, respectively. The diameter distributions of HH-NC and US-NC were 3 to 15 nm and 4 to 10 nm, respectively. The length and diameter distributions of HH-NC and US-NC were wider than those of AH-NC. Therefore, AH-NC had the smallest particles of the three preparation methods (Table 1). 

These results are comparable to those of other studies [2,34,35]. The average aspect ratios (L/D) of HH-NC and US-NC were 21.20 and 23.20, respectively, which were lower than that of AH-NC (36.73). According to a previous report [36], the reinforcing effect of NC is expected to improve as the aspect ratio increases, which may further improve the mechanical properties of biocomposites. The length and diameter of NC prepared from various sources has been reported: 100–300 nm length and 3–5 nm diameter from wood [37], 171.6 nm length and 14.6 nm diameter from cotton [38], 1160 nm length and 16 nm diameter from tunicate [39], and 100–1000 nm length and 10–50 nm diameter from bacterial cellulose [40]. 

The morphology and dimensions of NC depended on the preparation method and each method resulted in distinct features. Acid hydrolysis, a well-known method for preparing NC, results in a larger aspect ratio than the homogenization and ultrasonication methods. These mechanical process can break hydrogen bonds and disintegrate microfibers into nanofibrils, which form needle-like cellulose crystallites and consequently reduce the aspect ratio of NC.

### 3.5. XRD Analysis of MCC and NC

The effects of mechanical and chemical treatments on the crystalline structure of the cellulose samples were further characterized by XRD as shown in Figure 4. Both MCC and MD-MCC exhibited characteristic crystalline peaks near 2θ = 15.1°, 16.2°, 21.0°, 22.6°, and 34.5°; these peaks correspond to the (1$\overset{-}{1}$0), (110), (012), (200), and (004) crystallographic planes, respectively, and are characteristic of the cellulose I structure [41,42]. The AH-NC had the same crystalline peaks as MCC and MD-MCC except that the diffraction peak at 2θ = 34.5° was broader and flatter, and this peak disappeared after high-pressure homogenization and high-intensity ultrasonication. Generally, NC obtained via the three different preparation methods exhibited the characteristic cellulose I peaks, which indicates that acid hydrolysis, high-pressure homogenization, and high-intensity ultrasonication do not affect the main crystalline properties of cellulose. 

The crystallinity of NC is an important parameter as it determines its reinforcing capability and mechanical strength in composite films [43]. Highly crystalline fibers are expected to be more effective at providing reinforcement for composite materials because of their increased stiffness and rigidity, which result in a higher Young’s modulus. However, the crystallinity of MD-MCC and the prepared NC each decreased to a different extent relative to MCC (Table 2). The peak at 22.6° was less sharp for the alkali-treated MCC, which suggests that alkali swelling might destroy part of the crystalline structure [44]. Cationic modification of MCC disrupted both intermolecular and intramolecular hydrogen bonding and facilitated the formation of amorphous regions [6]. The results of this study indicate that the NC preparation methods were non-selective as they damaged both amorphous and crystalline cellulose; therefore, the crystallinity of NC decreased [2,7,45]. As shown in Table 2, the crystallinity of AH-NC (67.6%) was the highest of the three NC, which implies that the crystalline regions of MD-MCC were more resistant to the acid treatment than the mechanical treatments. Moreover, the crystallinity of US-NC (62.8%) was slightly higher than that of HH-NC (61.8%), which suggests that high-intensity ultrasonication was less aggressive than high-pressure homogenization.

### 3.6. Fourier Transform Infrared (FTIR) Spectroscopy Analysis of MCC and NC

Fourier transform infrared spectroscopy was used to understand the changes in the chemical structures of the MCC and NC (Figure 5). Two main absorption regions, 2800–3600 cm^−1^ and 750–1750 cm^−1^, were present in the spectra. The broad band centered at approximately 3332 cm^−1^ corresponds to O-H stretching of cellulose in the fiber [46]. The intensity of the O-H peak in the MD-MCC spectrum was considerably less than that of the MCC spectrum, which indicates that the number of hydroxyl groups in the MD-MCC was reduced by the etherification reaction [6]. The peaks near 2900 cm^−1^ are attributed to the C-H stretching vibration of cellulose and the peaks at 1644 cm^−1^ have been assigned to water absorption because of the strong cellulose-water interaction [47]. In addition to the characteristic peaks of the cellulose backbone, there was a small peak near 1479 cm^−1^ in the MD-MCC and NC spectra. This peak was not present in the MCC spectrum and thus was assigned to the trimethyl quaternary ammonium groups, which implies that trimethyl quaternary ammonium groups were successfully added on to cellulose chains even though the degree of substitution was low. 

The changes to the crystal structure of cellulose resulted in the intensity reduction of partial FTIR peaks attributed to the crystalline domains of cellulose [1]. The peak at 1427 cm^−1^ corresponds to the CH_2_ bending vibration of crystalline cellulose and the NC spectra exhibit reduced intensity of this crystalline band compared to those of the MCC. The reduced intensity of this peak also supports the decreased crystallinity of NC after acid hydrolysis, high-pressure homogenization, and high-intensity ultrasonication as demonstrated by the XRD results. The reduced crystallinity of NC compared to MCC implies that there is less intermolecular and intramolecular hydrogen bonding in NC, which might increase the dispersion of NC in water [1]. The peak at 1030 cm^−1^ is related to C-O stretching of the pyranose ring skeleton [48]. The peak at 890 cm^−1^ is corresponds to β-glycosidic linkages between the glucose units of cellulose. Based on the above analysis, there was not a significant difference between the MCC and NC spectra, which indicates that neither the mechanical nor chemical treatments changed the main chemical structure of the fibers. This result agrees with the XRD analysis.

### 3.7. Thermal Stability Analysis

The thermal stability of each of MCC, MD-MCC, AH-NC, HH-NC, and US-NC was investigated by TGA, as shown in Figure 6. The onset of thermal decomposition temperature (T_on_) corresponds to the beginning of degradation and the maximum decomposition temperature (T_max_) corresponds to the temperature of the maximum rate of degradation. The T_on_, T_max_, mass loss at T_max_, and the char residuals at 500 °C are given in Table 2. Generally, the thermal degradation of MCC and NC occurred in two steps, as shown in Figure 6. The initial mass loss, which was caused by the evaporation of absorbed water, was observed below 120 °C and was slightly different for the different cellulose samples.

As shown in Table 2, the T_on_ of each NC was less than that of MCC, which indicates that NC has decreased thermal stability. This decreased thermal stability might be attributed to the large number of cellulose chain segments that were damaged during the preparation of NC and formed low molecular chain segments and weak points in the cellulose chain on the surface of NC. At elevated temperatures, these low molecular chain segments and defects absorbed heat and thus began to degrade first, which resulted in the reduced thermal stability of NC. Furthermore, the nanoscale lateral dimensions of NC mean that NC has a higher surface to volume ratio than MCC and thus is heated more efficiently, which also decreases thermal stability [49]. Compared with AH-NC and US-NC, the lower thermal stability of HH-NC could be due to the increased damage to the crystalline region of cellulose during the high-pressure homogenization process as indicated by the XRD results [50].

The main thermal degradation stages of MCC, MD-MCC, AH-NC, HH-NC, and US-NC occurred in the range of 250–430 °C, 200–430 °C, 210–360 °C, 240–405 °C, and 250–405 °C, respectively; this degradation was mainly due to the thermal decomposition of the crystalline cellulose chains. The mass losses caused by the thermal decomposition of AH-NC, HH-NC, and US-NC at T_max_ were 50.8%, 52.9%, and 50.4%, respectively, and the corresponding T_max_ were 298 °C, 349 °C, and 365 °C. The mass losses of each type of NC were very similar but the corresponding T_max_ varied significantly, which indicates that acid hydrolysis, high-pressure homogenization, and high-intensity ultrasonication had different effects on the thermal stability of NC. The sulfate groups that were added to the cellulose chains during acid hydrolysis could facilitate the thermal degradation of cellulose [51], which would explain why the T_max_ of AH-NC was less than those of HH-NC and US-NC. Moreover, this result was consistent with the previous study by Wang et al. [36]. Above 450 °C, the thermal decomposition temperature of each cellulose sample leveled off and a slow thermal degradation profile was obtained. This continued degradation could be attributed to the carbonization of polysaccharide chains caused by the cleavage of C-C and C-H bonds.

The mass of char residue in AH-NC, HH-NC, and US-NC at 500 °C was 20.6%, 14.2%, and 8.5%, respectively, which are all greater than that of MCC (7.6%). The high char residue of AH-NC has been ascribed to the direct solid-to-gas phase transitions of decarboxylation catalyzed by the sulfate groups on the surface of AH-NC [52]. The high char yield of HH-NC might be due to its relatively high dehydration at low temperatures, which carbonized cellulose as confirmed by its relatively low degradation temperature. The low char residue yield of US-NC might be ascribed to its low crystalline content [2].

### 3.8. Mechanical Properties

The tensile strength (TS), elongation at break (EAB), and elastic modulus (EM) of starch films with and without the addition of 5% NC are shown in Table 3. TS is the measurement of maximum strength of a film against applied tensile stress, EAB represents the ability of a film to stretch, and EM indicates the rigidity of a film. The TS of the pure starch film was 6.32 MPa. The TS of the starch/NC films was considerably greater than that of the pure starch film. The TSs of the starch nanocomposite films prepared from AH-NC, HH-NC, and US-NC were 9.35 MPa, 11.74 MPa, and 10.75 MPa, respectively, which correspond to 1.47, 1.85, and 1.70 times the TS of pure starch film. Li et al. [22] reported that NC can improve the tensile strength of starch films when NC is uniformly distributed in the starch matrix. The strong adhesion at the starch/NC interface and the creation of a rigid NC percolating network within the starch matrix facilitate efficient stress transfer from the soft starch matrix to the rigid NC and thus improve the strength of the starch nanocomposite [53]. Moreover, cationic modification is a hydrophilic modification that introduces hydrophilic quaternary ammonium salt groups; therefore, the cationic NC has excellent compatibility and strong interactions with starch because hydrogen bonds can form between them, which greatly increases the TS of the nanocomposite films.

The reinforcing effect of NC on the starch films is also different for the different preparation methods. This is potentially due to the different aspect ratios of the three types of NC, which could affect the dispersion of NC in the starch matrix [6]. Since the AH-NC has the highest aspect ratio, hydrogen bonding might prevent it from being uniformly dispersed throughout the starch matrix [28]. Conversely, HH-NC has a low aspect ratio and thus is expected to be well-dispersed in the starch film. This conjecture was verified using the SEM results of the films, which are discussed later in this section. 

The effects of preparation method of NC on the EAB of starch/NC composite films were shown in Table 3. As expected, the incorporation of NC into the starch matrix substantially decreased the EAB and the results were in agreement with previous studies [54,55]. The decrease in EAB is attributed to the geometry and rigid nature of NC as well as the formation of a stiff NC network linked by hydrogen bonds and entanglements. Moreover, the addition of NC could hinder the plasticizing efficacy of glycerol and decrease the mobility of starch chains, which would result in brittle nanocomposite films [56,57]. Effects similar to those observed for NC have been reported for other biopolymers, such as whey protein isolate and agar [42,58]. Finally, EM, which indicates the rigidity of a film, increased significantly (*p* < 0.05) when 5% NC was added to the starch film.

### 3.9. Water Vapor Barrier Properties 

The water vapor permeability (WVP) of a film is one of the most important properties for food packaging applications. Because of its hydrophilic nature, starch-based films usually have high WVP and thus poor water vapor barrier properties, which significantly restricts its application in food packaging. As shown in Table 3, the water vapor barrier properties of pure starch films could be improved by the incorporation of NC. The WVP of the pure starch film, 1.65 × 10^−12^ g·cm·cm^−2^·s^−1^·Pa^−1^, decreased by approximately 14%, 22%, and 18% (*p* < 0.05) when 5% of AH-NC, HH-NC, and US-NC was incorporated into the starch matrix, respectively. This decrease in WVP as the result of the incorporation of NC agrees with previous reports [22,59]. 

The NC in the starch nanocomposite films functions as an impermeable barrier against WVP because strong hydrogen bonding interactions reduce the diffusion coefficient of the films by increasing the diffusion path for water vapor through the film [42]. Unexpectedly, the lower the aspect ratio of the NC in the starch matrix, the lower the WVP of the resultant starch nanocomposite films. This result might be ascribed to the agglomeration of high aspect ratio NC in the starch matrix, which creates diffusion pathways for water vapor transport within the film and thus facilitates WVP. Therefore, the uniform distribution of NC in the starch matrix played a more significant role than high aspect ratio in improving the water vapor barrier properties of starch nanocomposite films. 

### 3.10. Surface Morphology

The surface morphologies of pure starch film and starch nanocomposite films with 5% NC are shown in Figure 7 at 10,000× magnification. The surface of the pure starch film was smooth with homogeneous morphology and compact structure because of the plasticization effect of glycerol [60]. Compared to the pure starch film, the incorporation of AH-NC increased the surface roughness, which is attributed to the agglomeration of AH-NC in the starch matrix [22]; however, it was difficult to observe the individual fibers in the film because of the small size and bundle structure of AH-NC in the starch matrix. Well-dispersed NC particles were observed in the starch/HH-NC and starch/US-NC films; this suggests that there was strong interfacial adhesion or good compatibility between the HH-NC or US-NC and the starch matrix, which could be attributed to the size and zeta potential of the NC, the chemical similarities of starch and NC, and hydrogen bonding between the two components [22]. Similar results have been reported for starch nanocomposite films containing pineapple leaf cellulose nanofibers [61]. The homogeneous distribution of NC in starch films could greatly improve their tensile strength and water vapor barrier properties.

## 4. Conclusions

High yields of NC with different aspect ratios were obtained efficiently from cationic MD-MCC by acid hydrolysis, high-pressure homogenization, and high-intensity ultrasonication and the effects of AH-NC, HH-NC, and US-NC on the fundamental properties of starch films were compared. The cationic modification of NC was confirmed by FTIR analysis. Both HH-NC and US-NC prepared from MD-MCC were stable suspensions because of their higher zeta potential compared to NC samples prepared from MCC. The AH-NC prepared from MD-MCC tended to aggregate because of the presence of sulfate groups and, thus, lack of surface charge. The XRD analysis revealed that the crystallinity of NC decreased for all preparation methods whereas the main chemical structure of fibers remained unchanged. The thermostability of the three types of NC decreased relative to that of MCC. The HH-NC exhibited the best dispersion in the starch matrix and demonstrated the best enhancement to the water vapor barrier properties of starch films. Cationic modification of MCC will be a promising strategy to improve the yields and dispersion of NC and ultimately enhance the properties of starch films. The starch/NC nanocomposite films, which are completely biodegradable and biocompatible, have immense potential for food-packaging applications.

## Figures and Tables

**Figure 1 nanomaterials-09-01702-f001:**
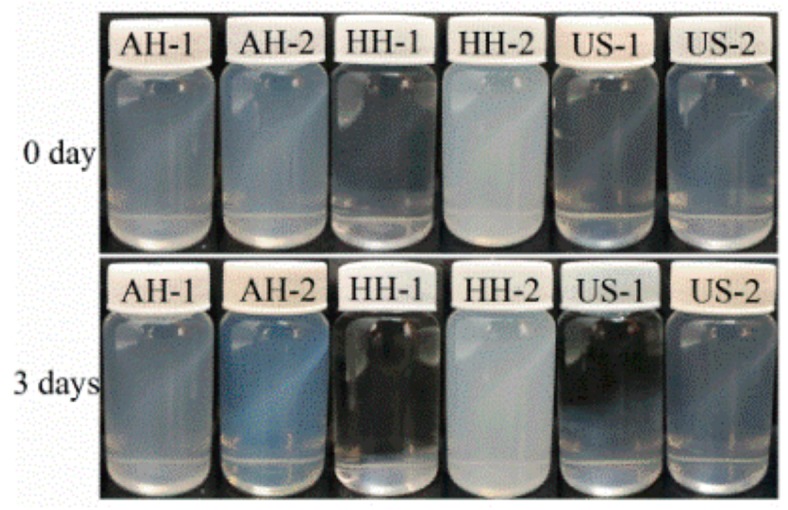
The suspension stability of NC prepared from MCC (AH-1, HH-1, US-1) and MD-MCC (AH-2, HH-2, US-2) t = 0 day and t = 3 days.

**Figure 2 nanomaterials-09-01702-f002:**
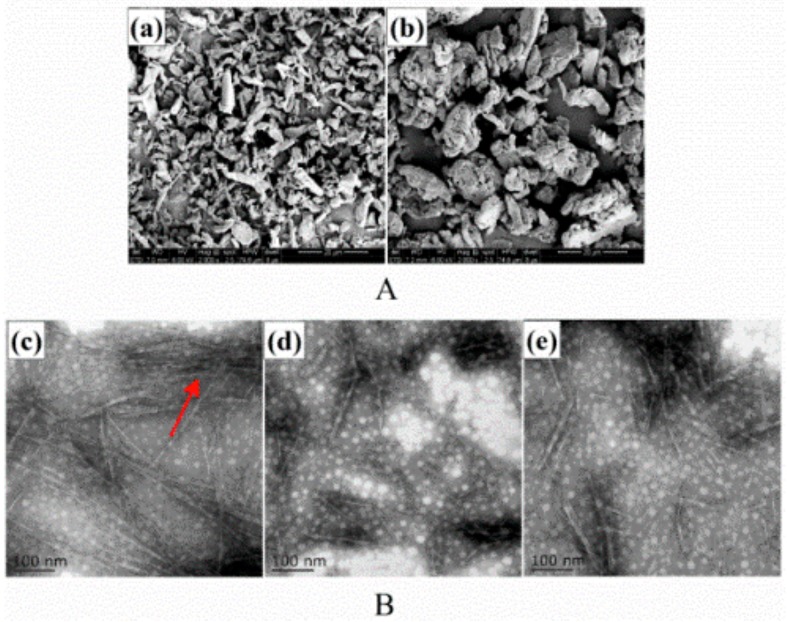
(**A**) Scanning electron microscopy (SEM) micrographs of (**a**) MCC and (**b**) MD-MCC; (**B**) transmission electron microscopy (TEM) micrographs of (**c**) AH-NC, (**d**) HH-NC, and (**e**) US-NC.

**Figure 3 nanomaterials-09-01702-f003:**
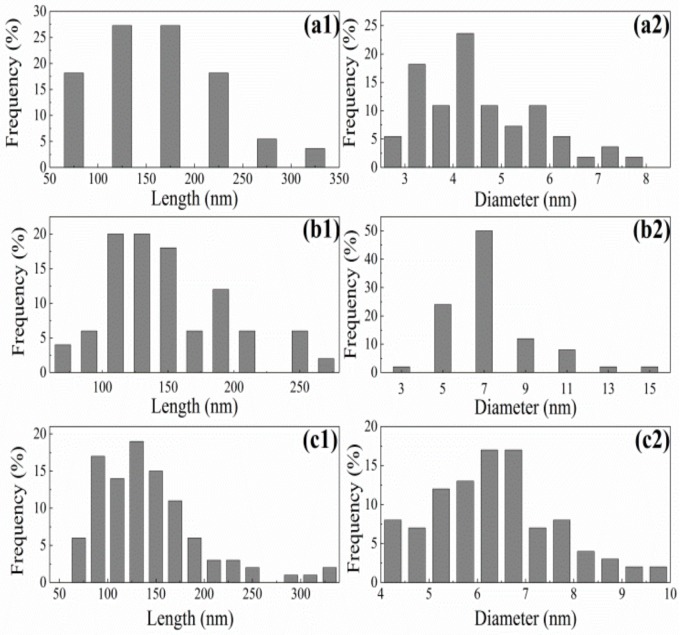
Length-frequency and diameter-frequency histograms of (**a**) AH-NC, (**b**) HH-NC, and (**c**) US-NC.

**Figure 4 nanomaterials-09-01702-f004:**
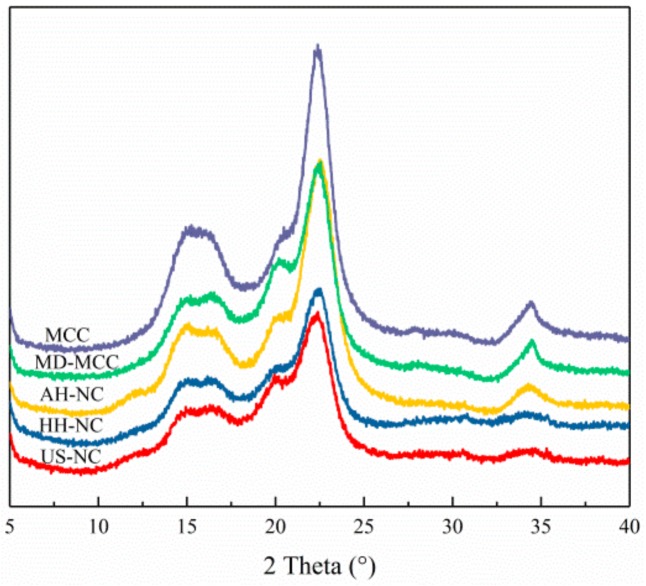
X-ray diffraction (XRD) patterns of MCC, MD-MCC, AH-NC, HH-NC, and US-NC.

**Figure 5 nanomaterials-09-01702-f005:**
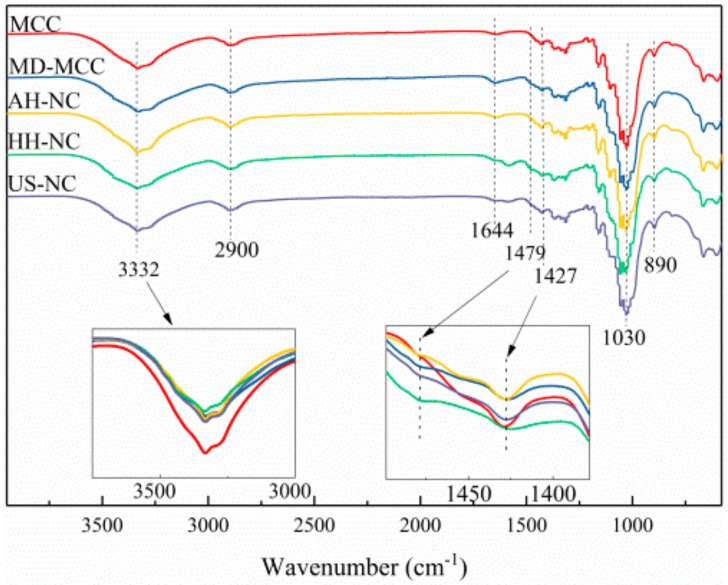
Fourier transform infrared (FTIR) spectra of MCC, MD-MCC, AH-NC, HH-NC, and US-NC.

**Figure 6 nanomaterials-09-01702-f006:**
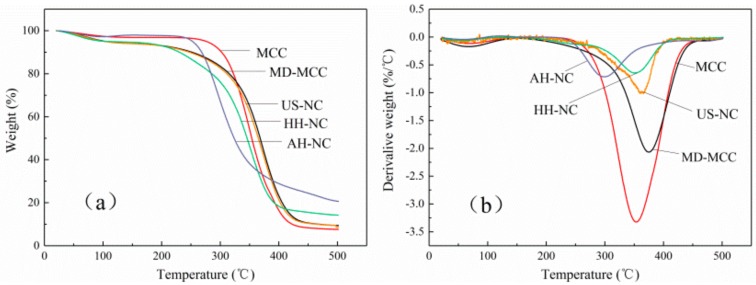
(**a**) Thermogravimetric analysis (TGA) and (**b**) derivative thermogravimetry (DTG) curves of MCC, MD-MCC, AH-NC, HH-NC, and US-NC.

**Figure 7 nanomaterials-09-01702-f007:**
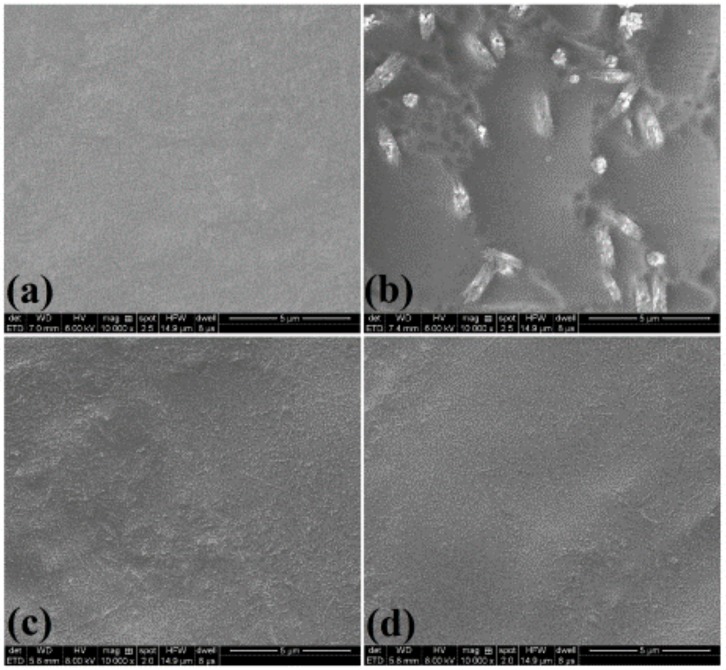
Surface morphology of (**a**) pure starch, (**b**) starch/AH-NC, (**c**) starch/HH-NC, and (**d**) starch/US-NC films.

**Table 1 nanomaterials-09-01702-t001:** Effect of preparation method on the yield, zeta potential, and average particle size of NC prepared from MCC and MD-MCC.

	NC Samples	Raw Materials	
		MCC	MD-MCC
**Yield (%)**	AH-NC	20.68 ± 0.04	30.63 ± 0.08
	HH-NC	2.04 ± 0.14	33.08 ± 0.34
	US-NC	3.57 ± 0.21	14.18 ± 0.10
Zeta potential (mV)	AH-NC	−5.82 ± 0.09	−5.17 ± 1.18
	HH-NC	−0.71 ± 1.02	15.66 ± 0.44
	US-NC	−1.41 ± 0.85	16.01 ± 0.66
Average particle size (nm)	AH-NC	69.5 ± 0.3	69.0 ± 0.3
	HH-NC	253.3 ± 0.4	161.4 ± 1.4
	US-NC	347.9 ± 5.6	255.0 ± 2.9

**Table 2 nanomaterials-09-01702-t002:** Crystallinity index (CI), degradation temperature, and mass loss of MCC, MD-MCC, AH-NC, HH-NC, and US-NC.

Samples	CI (%)	T_on_ (°C)	T_max_ (°C)	Weight loss at T_max_ (%)	Residue at 500 °C (%)
MCC	71.4	286	353	49.4	7.6
MD-MCC	68.2	278	374	56.6	9.4
AH-NC	67.6	249	298	50.8	20.6
HH-NC	61.8	228	349	52.9	14.2
US-NC	62.8	279	365	50.4	8.5

**Table 3 nanomaterials-09-01702-t003:** Mechanical properties and water vapor permeability of starch/NC composite films.

Samples	TS (MPa) *	EAB (%) *	EM (MPa) *	WVP (10^−12^ g·cm·cm^−2^·s^−1^·Pa^−1^) ^*^
starch	6.32 ± 0.86d	25.33 ± 0.85a	369.51 ± 13.67c	1.65 ± 0.12a
starch/AH-NC	9.35 ± 0.90c	20.1 ± 0.61b	492.69 ± 10.37b	1.42 ± 0.06b
starch/HH-NC	11.74 ± 0.52a	18.8 ± 0.83b	573.08 ± 13.24a	1.28 ± 0.04b
starch/US-NC	10.75 ± 0.79b	19.6 ± 0.76b	513.24 ± 24.25b	1.36 ± 0.88b

* Different lowercase letters in the same column indicate a statistically significant difference (*p* < 0.05).

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
