# Peer review of "Effects of Preparation Method on the Physicochemical Properties of Cationic Nanocellulose and Starch Nanocomposites"

_nanomaterials, 2019, doi:10.3390/nano9121702_

Round 1

Reviewer 1 Report

This paper (nanomaterials-643675) is a report regarding the effects of preparation method on the physicochemical properties of cationic nanocellulose and starch nanocomposites. The use of sodium hydroxide solution treatment to make NC and CNF is widely known (*). It is also common to use high pressure homogenizers or ultrasonic illumination to make NCs. The novelty of this paper should be explicitly discussed. *e.g:K. Abe et. al. Cellulose, 23, 1257 (2016)

Author Response

Response to Reviewer 1 Comments

Dear Reviewer:

We are grateful to the reviewer’s suggestions. We have already revised the manuscript according to these suggestions point by point.

Point 1: This paper (nanomaterials-643675) is a report regarding the effects of preparation method on the physicochemical properties of cationic nanocellulose and starch nanocomposites. The use of sodium hydroxide solution treatment to make NC and CNF is widely known (*). It is also common to use high pressure homogenizers or ultrasonic illumination to make NCs. The novelty of this paper should be explicitly discussed. *e.g:K. Abe et. al. Cellulose, 23, 1257 (2016)

Response 1: Thank you very much for your comments and suggestions. The novelty of this paper had been explicitly discussed. Most of the previous studies modified the NC directly. However, the nanostructure of NC might be destroyed because of the alkalization process when the NC was modified by cation etherification directly. Therefore, it was difficult to prepare NC with modified group. However, alkali-swelling can disrupt hydrogen bonding between microfibrils in pulp fibers and could facilitate mechanical nanofibrillation. In this paper, we adopted a novelty method that MCC was modified with cation etherification modification at first. And then, cationic NCs were prepared from the cationic-modified MCC by acid hydrolysis, high-pressure homogenization, and high-intensity ultrasonication, respectively. Moreover, we found that the yields of NCs prepared from cationic-modified MCC were higher than that of MCC. Because alkali-swelling can disrupt hydrogen bonding between microfibrils in pulp fibers and could facilitate mechanical nanofibrillation. It is positive significance for resource conservation and environmental protection.

Reviewer 2 Report

The author reported an investigation of the effects of preparation methods on the properties of NC. Although the results are convincing and surface characterization techniques are well selected, however, there are a few changes that should be taken into account before considering it for publication.

Line 259: “chemical structure” is not 100% accurate as the paragraph discussed the XRD results which mainly related to crystal structure. Therefore, the main “crystalline properties” or main “crystal structure” would suit more. It would be great to rewrite the last sentence in line 302 accordingly, as well as the corresponding sentence in the Conclusion.

Line 391: “the incorporation of AH-NC greatly increased the surface roughness”. This statement also not 100% correct. It would be better to perform AFM or a profilometry measurement to make it clear how surface roughness is changed with surface modification.

I strongly recommend rewriting these statements which considered either confusing or unclear, sometimes too much information in a single sentence:

Line 163: The yields of AH-NC prepared from MCC and MD-MCC were 20.68%

and 30.63%, respectively, which indicates that cationic modification improved NC yield by 48.1%.

Line 232: The approximate length and diameter distributions of HH-NC were 25 to 275 nm and 3 to 15 nm with maxima 232 at 125–150 nm and 7 nm, respectively.

Suggestion: Use the same expression as line 242.

233: The length and diameter distributions of US-NC were 75 to 350 nm and 233 4 to 10 nm with maxima at 80–150 nm and 6–7 nm, respectively.

My suggestion which is not obligatory for this manuscript: Performing the XPS would enhance the study.

Author Response

Response to Reviewer 2 Comments

Dear Reviewer:

We are grateful to the reviewer’s suggestions. We have already revised the manuscript according to these suggestions point by point.

Point 1: Line 259: “chemical structure” is not 100% accurate as the paragraph discussed the XRD results which mainly related to crystal structure. Therefore, the main “crystalline properties” or main “crystal structure” would suit more.

Response 1: The paper has been revised according to the above suggestion.

Point 2: It would be great to rewrite the last sentence in line 302 accordingly, as well as the corresponding sentence in the Conclusion.

Response 2: We have rewritten the last sentence in line 302, but the Conclusion doesn't involve corresponding sentence.

Point 3: Line 391: “the incorporation of AH-NC greatly increased the surface roughness”. This statement also not 100% correct. It would be better to perform AFM or a profilometry measurement to make it clear how surface roughness is changed with surface modification.

Response 3: Thank the reviewer for these comments and suggestions. However, in the present study, SEM of the films can indicate that the surface of the film became rough after adding AH-NC, compared with that of  pure starch film. Li et al. [1] observed the incorporation of cellulose nanofibers greatly increased the surface roughness of the composite films compared to that of the starch-only film by SEM. Wang et al. [2] reported that the surface morphology of pure agar film was smooth and homogenous, but some agglomerations were observed with the nano-bacterial cellulose concentrations increased using SEM.

References

[1] Li M.; Tian X.; Jin R.; Li D. Preparation and characterization of nanocomposite films containing starch and cellulose nanofibers, Ind. Crop. Prod. 2018, 123, 654-660.

[2] Wang X.J.; Guo C.F.; Hao W.H.; Ullah N.; Chen L.; Li Z.X.; Feng X.C. Development and characterization of agar-based edible films reinforced with nano-bacterial cellulose, Int. J. Biol. Macromol. 2018, 118, 722-730.

Point 4: I strongly recommend rewriting these statements which considered either confusing or unclear, sometimes too much information in a single sentence:

Line 163: The yields of AH-NC prepared from MCC and MD-MCC were 20.68% and 30.63%, respectively, which indicates that cationic modification improved NC yield by 48.1%.

Line 232: The approximate length and diameter distributions of HH-NC were 25 to 275 nm and 3 to 15 nm with maxima 232 at 125–150 nm and 7 nm, respectively.

Response 4: The two sentences have been revised according to the above suggestion.

Point 5: My suggestion which is not obligatory for this manuscript: Performing the XPS would enhance the study.

Response 5: Thank you for your suggestion. In the present study, we think that the experimental results may not be optimal, but should be sufficient. We will consider using XPS in the next study. Thank you again.

Reviewer 3 Report

The authors present a manuscript detailing the production of nanocrystalline cellulose by different methods.The manuscript is very well written and readily understood. The language used is very good. Overall the presented research could be valuable to other researchers in the field of cellulose composite materials.  The researchers presented a good introduction of the problem, and also they provided a good background of previous studies. The researchers were very careful in describing the experimental part. Therefore, these experiments could be replicated by other researchers. They provide strong characterization of the materials with FTIR, XRD, SEM,TGA, zeta potential, mechanical testing, thus giving an insight into the physical and chemical structure of the produced materials. The authors detailed the importance of pretreatment and any effect that could have on the nanocellulose. The conclusions in this manuscript are supported by the experimental results. Thus I support the publication of this manuscript.

Author Response

Response to Reviewer 3 Comments

Dear Reviewer:

Thank you very much for your appreciation and great efforts on our manuscript.

Reviewer 4 Report

This manuscript focuses on practical approaches. First, experiments are described that isolate cationized nanocellulose by either of three methods (acid hydrolysis, high-pressure homogenization, and high-intensity homogenization) from a commercially available material that is enriched in cellulose. Secondly, these methods have been compared with respect to parameters, such as yield, Zeta potentials, average size, and crystallinity of the nanocellulose. Finally, the effect of the various nanocellulose preparations on physico-chemical propertise of starch nanocomposites has  been determined. It seems that the experiments and their presentation in the manuscript to be reviewed follow a clear structure. The Content of this manuscript is expected to be of interest to readers of the journal. The language, however, should be checked.

In addition, I have only a few minor comments:

Presumably, it would help Readers if some terms (such as Young's modulus - line 34 pg 2 and TEMPO oxidation - line 37 pg 3) would briefly explained. To achieve a higher degree of homogeneity throughout the manuscript, the % value of the sulfuric acid should be defined (w/v or v/v).  Line 91 pg. 4: The period of time should be mentioned in order to help reproducibility of the results.  Sometimes, the text can be shortened. E.g. would it not be possible to write 'According to Wang et al. (50) ...' (line 329 pg. 13)?

Author Response

Response to Reviewer 4 Comments

Dear Reviewer:

We are grateful to the reviewer’s suggestions. We have already revised the manuscript according to these suggestions point by point.

Point 1: Presumably, it would help Readers if some terms (such as Young's modulus - line 34 pg 2 and TEMPO oxidation - line 37 pg 3) would briefly explained.

Response 1: The paper has been revised according to the above suggestion (Young's modulus - line 348 and TEMPO oxidation – line 37).

Young's modulus is a kind of elastic modulus, which indicates the rigidity of a film. TEMPO is 2,2,6,6-tetramethylpiperidine-1-oxyl radical. TEMTO oxidation is a modification method which use 2,2,6,6-tetramethylpiperidine-1-oxyl radical to modify nanocellulose.

Point 2: To achieve a higher degree of homogeneity throughout the manuscript, the % value of the sulfuric acid should be defined (w/v or v/v).

Response 2: The paper has been revised according to the above suggestion.

Point 3: Line 91 pg. 4: The period of time should be mentioned in order to help reproducibility of the results.

Response 3: The high-pressure homogenization method can achieve reproducibility of the results  by determining homogenization times. A gram of MCC/MD-MCC was added to 100 mL of deionized water and 4 times at 800 bar using a SCIENTZ-150 high pressure homogenizer.

Point 4: Sometimes, the text can be shortened. E.g. would it not be possible to write 'According to Wang et al. (50) . (line 329 pg. 13)

Response 4: The paper has been revised according to the above suggestion.